# Processes of Sub-Citizenship: Neoliberal Statecrafting 'Citizens,' 'Non-Citizens,' and Detainable 'Others'

**Daile Lynn Rung**

The Northern Institute, Charles Darwin University, Darwin 0810, Australia; dailelynn.rung@cdu.edu.au

**Abstract:** Increasingly, scholars are exploring the politics of migration and the shifting terrain of citizenship from a critical mobilities perspective. To contribute to these discussions, in this paper, I explore how processes of sub-citizenship occur as nation-states craft immigration, citizenship, and border securitization policies and practices. I argue that complex and shifting processes of sub-citizenship largely occur through the nation-state's production of 'insiders' ('citizens') and 'outsiders' ('non-citizens'). As a nascent attempt to introduce sub-citizenship, I draw upon recent high-profile cases of family separation, abuse, and neglect experienced by children with 'illegal migrant' status in the United States and Australia. Under the international nation-state system and the neoliberal globalization paradigm, the border policing powers of nation-states are primed to expand and intensify processes of sub-citizenship. Those at lower levels of the sub-citizen hierarchy are at risk of experiencing various forms of state-led violence, including deportation, detention, and torture.

**Keywords:** human mobility; citizenship; children in detention; nation-state; border policing; illegalization; neoliberalism; USA; Australia; immigration detention

## 1. Introduction

> It is not always the case that the governing political philosophy is spoken by everybody as if they're already inside it. It is when it becomes 'just how things are' that it wins consent and enters common sense. And at that point the political regime or philosophy has achieved a more settled, long-term deeper form of control . . . a level of unconsciousness where people aren't even aware they're speaking ideology at all. The ideology has become 'naturalised,' simply part of nature. (Hall and Massey 2010, p. 59)

There is little doubt that the international system of nation-states and the closely related concept of citizenship form the governing political philosophy of the modern era. The hegemony of nationalist thought and institutions is evidenced by how the international system of nation-states is popularly regarded as being common sense and the natural order of things. Rather than passively accepting the international system of nation-states and the closely connected migration, citizenship, and border regimes as 'just how things are,' there is a pressing need for research to (1) examine how nation-states' policies, practices and discourses relating to immigration, citizenship, and border securitization actually affect the everyday lives of people around the world; (2) interrogate and historicize these ideological constructs; and (3) imagine alternatives to the current globalized system of international nation-states, securitized citizenship, and militarized border regimes.

This article is my first attempt to develop a new theoretical concept I call 'processes of sub-citizenship.' I define processes of sub-citizenship as various forms of social, political, economic, and

territorial expulsion[1] that create 'sliding scales of subordination' for people with different citizenship and migrant statuses (Sparke 2016). Sub-citizenship is produced by unequal relations between people construed by nation-states as belonging to one of two constructed binaries: 'citizens' and 'non-citizens.'[2] It is a relational process that creates and justifies social boundaries between people with 'citizen' and 'non-citizen' statuses in evolving and complex ways. Those with 'illegalized'[3] migrant and citizen statuses are particularly affected by these processes that create and reproduce social boundaries and inequality. My intention for this article is to apply sub-citizenship as a heuristic device to understand how translocal processes of subordination create various hierarchical conditions of precarity and dehumanization for different groups of people based upon, but not wholly determined by, migration and citizenship status.

This theoretically-oriented paper has two main objectives. First, it offers a new way to understand the ordering of populations and the violence that is often connected to categorizations of personhood along neoliberal lines. Second, it draws upon recent high-profile cases of children in immigration detention in Australia and America to explore how processes of sub-citizenship may affect the lives of people located at the lower levels of the hierarchy. To explore processes of sub-citizenship, I employ a qualitative method of online content analysis of relevant immigration detention practices in Australia and the United States. To identify and contextualize the human costs associated with processes of sub-citizenship in these two countries, I include relevant gray literature with reports and newspaper articles relevant to children's experiences in immigration detention.

I put forth the following thesis: the logic of neoliberal reengineering, which is chiefly carried out by territorially bounded nation-states located within the international nation-state system, produces conditions of sub-citizenship for most people regardless of their citizenship status. The institutional machinery involved in producing and categorizing insiders ('citizens') and outsiders ('non-citizens') is structurally connected to processes of sub-citizenship for all. However, people construed by nation-states as being 'illegal migrants' status are particularly vulnerable to processes of sub-citizenship.

The neoliberal market paradigm is the driving force behind process of sub-citizenship. As the nation-state assumes its role as the core agency that redraws the tenor of citizenship (Wacquant 2012), processes of sub-citizenship increasingly affect those who are internationally mobile, but also affect those who are not. To be clear, migrating across borders does not produce sub-citizenship. Rather, processes of sub-citizenship are intimately linked to "the state and social inequality in the bloom of neoliberalism" (Wacquant 2012). Sub-citizenship is largely produced through the categorization, or the potential categorization, of less secure citizen and migrant statuses, such as 'non-citizen,' 'temporary migrant,' and 'unlawful non-citizen.' Processes of sub-citizenship are highly differentiated among 'citizens' and 'non-citizens.' However, it must be understood that sub-citizenship occurs through the autonomy of the neoliberal state, where increasingly even the most basic services, safety nets, and human rights can only be accessed by those who are considered 'deserving' based their ability to work and/or have access to capital (Brodie 1997; Grace et al. 2017; Sagy 2013; Somers 2008).

---

[1]  I use the term 'expulsion' to refer to a qualitative "deprivation of status" (Nail 2016). It can take many forms, including removing social, political, juridical, economic, and territorial status from a person or a group of people. For example, certain groups of people with legalized 'non-citizen' status, such as 'temporary migrants' and 'permanent residents,' often do not have (or have limited access to) voting rights, welfare provisions, public education, work rights, and so on. People with illegalized 'non-citizen' status are at risk of experiencing more severe forms of social, political, juridical, economic, and territorial expulsion, including deportation and detention.

[2]  The Office of the United Nations High Commissioner for Human Rights defines a 'non-citizen' as a person who has not been recognised as having "effective links to the country where he or she is located" (Office of the United Nations High Commissioner for Human Rights 2006, p. 5). In the United States, 'alien,' is the more commonly used term to refer to people without citizenship status. (Immigration and Nationality Act 1952). In the interest of consistency, this paper will use the term 'non-citizen.'

[3]  The term 'illegalized' is used to convey "that states of illegality are not in any way natural, but are deliberately constructed through law and practice" (Weber 2013).

To examine the politics of human mobility and the human costs of managerialist approaches to migration, we must take on a more critical, interrogative stance with respect to migration and citizenship statuses that are applied to different groups of people. In this article, I put the person before the migration and/or citizenship status or put the designated status in scare quotes. My intention is to disrupt and denaturalize migration and/or citizenship statuses that place people into different legal and social classes. In practice, this entails describing people not as being illegal migrants, citizens, non-citizens, and so forth who experience X, Y, and Z. Rather than taking migration and citizenship statuses as though they are natural, obvious, or inherent characteristics of people, we can talk in terms of people who are construed, or imagined, as being members of various groups. By focusing on the current power dynamics operating within a nation-state we can observe the processes by which certain people are categorized into legal statuses including 'the citizen,' 'the non-citizen,' 'the permanent resident,' 'the temporary migrant,' 'the illegal migrant,' and so on. Exploring the political crafting of citizenship and migrant statuses through an historical lens allows us to piece together how these statues situate people to have diverse, and highly uneven, experiences accessing human mobility and human rights. This is an important analytic and discursive tactic for elucidating the politics underpinning migration and the selective processes that nation-states and other actors have in allocating access to mobility and human rights among different segments of the population. Focusing on the political organization and historical evolution structuring migrant and citizenship status opens analytic and methodological spaces to explore and map the processes underpinning how it has come to be that certain people experience different levels of subordination and dehumanization and what role citizenship and migration statuses play in these historically, politically and socially constructed dynamics.

When 'migrant' and related terms[4] appear in this paper, they are used to refer to people who move and resettle transnationally (that is, across international borders). Migrants include people whose movement across borders occur on a spectrum with various degrees of choice, and whose movement or presence is interpellated by the nation-state as being 'legal' or 'illegal.' All migrants, at least for a time, acquire the status of 'non-citizen.' However, moving across international borders is not the only way for people to acquire 'non-citizen' status.[5] At times, I purposefully select the term 'non-citizen' to draw attention to people's citizenship status within the nation-state. Highlighting legal status is important for investigating how different institutionalized categories affect the everyday lives of people with different legal citizenship and migrant statuses regardless of how long they may have physically lived within the state's territorial borders.

Research in the new mobilities approach explores how human life is organized and shaped dynamically across local and national boundaries. The new mobilities paradigm views all social phenomena, including human mobility, through a 'mobile sociology' (Sheller 2011) that rejects a 'sedentarist metaphysics' (Malkki 1992) in favor of a 'nomadic' one (Cresswell 2006). This paper draws upon the analytical and methodological toolkits offered by a mobilities approach along with the insights of No Borders Studies. A No Borders perspective asserts that people should not be sorted, labeled and categorized through inherently exclusionary state forms of identification such as 'migrant' and 'citizen' (Baines and Sharma 2002; Raithelhuber et al. 2018). This paper applies sub-citizenship as a new theoretical lens to exploration how human life is organized within contemporary neoliberal processes.

---

4    In this paper, I use the term 'migrant' to denote a person who has moved across international borders. Terms related to 'migrant' include 'forced migrant' (including 'refugee' and 'asylum seeker'), 'illegal migrant,' 'undocumented migrant,' 'voluntary migrant,' 'temporary migrant,' 'economic migrant,' 'internal migrant,' and so on. These terms either represent legal statuses or are terms of rhetoric. For instance, 'illegal migrant' is a form of rhetoric denoting people who do not have the nation-state's authorization to be within its territorial borders. At times, 'illegal migrants' are referred to as 'undocumented migrants,' which highlights that they do not have the nation-state's permission to move.

5    The other ways that people can become 'non-citizens' include being born in a country without citizenship status, opting to be denaturalized, or having one's citizenship stripped. It is beyond the scope of the present article to explore the various ways that people may acquire 'non-citizen' status and their operant effects.

I draw upon recent high-profile cases of children in immigration detention to illustrate the human cost of processes of sub-citizenship among those located at the bottom of the sub-citizen hierarchy.

The structure of the article is as follows. First, I briefly discuss of how citizenship is reengineered through "neoliberal statecraft" (Wacquant 2010, 2012). Second, I introduce processes of sub-citizenship as a new theoretical construct for exploring the structural and ideological conditions shaping people's experiences with (im)mobility and access to human rights. Third, to illustrate how processes of sub-citizenship may affect the everyday lives of certain groups of people, I draw upon examples of contemporary human rights abuses experienced by children who are construed as having 'illegal migrant' status in Australia and the United States. Fourth, I take a step back from the present to historicize and denaturalize the ideological construct of the international nation-state system itself. Finally, I offer a brief preliminary sketch of how adopting a sub-citizenship lens to investigate human (im)mobilities, human rights abuses, and the more diverse and subtle forms of precarity experienced by people with different citizenship and migrant statuses offers the potential to mobilize radical new forms of solidarity that could lead to entirely new mobility futures.

## 2. 'Reengineering' Citizenship

Citizenship is not static; rather, it is a dynamic and socially determined process. Because it is socially determined and because, in practice, "one has to be a citizen of a state that is willing and able to protect her rights in order to have human rights" (Sagy 2013, p. 232), citizenship is also a highly contested status (Lister 2007). Modern citizenship is a contract-like arrangement that gives rise to social boundaries (Tilly 2005). There has been a long history of struggle over the policies and practices used to decide the line between citizenship and various lesser statuses (De Genova 2007; Hindess 2000; Isin 2002; Papadopoulos and Tsianos 2013; Tilly 2005; Turner 2007). Formations of citizenship are the result of social struggles and bargaining negotiations that rarely take place between equals (Tilly 2005). Citizenship has been described as a 'momentum concept' that must be continuously revisited, reworked, and refined to realize more of its egalitarian and anti-hierarchical potential (Lister 2007).

In early and in later phases of nation building nation-states created insider ('citizen') and outsider ('non-citizen') identities. To acquire things like political power, military might, and human capital nation-states bargained citizenship to populations (Tilly 2005). In the modern era citizenship and migrant statuses continue to be bargained to certain populations. However, modern-day citizenship has shifted towards 'neoliberal citizenship' (Ong 2007; Petcu 2015; Sparke 2004, 2016; van Houdt et al. 2011; Wacquant 2010, 2012), where citizenship has been reframed in an increasingly contractual and contingent manner. The shift towards neoliberal citizenship has gone hand and hand with 'market citizenship' (Brodie 1997; Grace et al. 2017), where the allocation of citizenship rights is increasingly based on an individual's economic power and participation in the labor market.

In more recent years, neoliberal capitalist economies have evolved to "become structurally dependent on the availability and continual supply of migrants' labor" (Bauder 2008, p. 316). However, nation-states, particularly those in the more affluent West, tend to gloss over their structural dependency on cheap, exploitable migrant labor and the various revenue streams associated with migration. Thomas Nail's notion of the "indefinite labor circuit" (Nail 2015, p. 32) elaborates how capitalist expansion relies upon the ability to "indefinitely extend the extraction of movement from the migrant flow and harness it into the many junctions of the economy" (p. 32). Mobility controls regulating and policing people's movements within and between nation-states reproduce a global economy of "disempowered migrant labor that props up the empowered labor and wages of citizens" (p. 32). In recent years, migration is increasingly framed in term of 'crisis' with nation-states enacting ever more militarized control methods, which are increasingly privatized and profitable (Castles 2010; McNevin 2007; Weber 2013).

It is the infrastructures and concentrations of mobile capital that "at one and the same time enhance the potential mobility for some, while detracting the potential mobility . . . of others by leaving them in a relatively slower or *intentionally disconnected* position" (Sheller 2011, p. 4; my emphasis).

Intentionally manipulating human flows to accelerate, decelerate, connect, and disconnect people from accessing mobility are 'spatial fixes' (Harvey 2001) and 'spatio-temporal fixes' (Jessop 2006). By putting in place various forms of mobility controls, nation-states participate in "capitalism's insatiable drive to resolve its inner crises tendencies by geographical expansion and geographical restructuring" (Harvey 2001, p. 24).

Neoliberal statecraft, which has come to prominence since the 1980s, has recast citizenship in such a manner that it is no longer regarded as "a prima facie right but as a prized possession that is to be earned and can be lost if not properly cultivated" (van Houdt et al. 2011, p. 408). Neoliberal statecraft has a distinct and recognizable institutional core that "consists of an *articulation of state, market and citizenship* that harnesses the first to impost the stamp of the second onto the third" (Wacquant 2012, p. 71). While 'neoliberalism has always been an open-ended, plural and adaptable project" (Peck 2008), scholarship that claims 'there is no such thing as neoliberalism' (Barnett 2005) or conceptualises it as a 'necessary illusion' (Castree 2006) fails to link local expressions of violence to larger scale political and economic processes. By theorizing neoliberalism as an 'actually existing circumstance' (Brenner and Theodore 2002), this paper contributes to identifying and understanding the nonillusory effects of neoliberalism (Springer 2008).

Loïc Wacquant's "'thick' sociological conception" of neoliberalism centered on the state is helpful to understand how citizenship is linked to the four key elements of "neoliberal reengineering" (Wacquant 2010, 2012), which includes the expansion of (1) commodification; (2) disciplinary social policy; (3) penal policy; and (4) the troupe of individual responsibility. These four elements specify "the institutional machinery involved in the establishment of market dominance and its operant impact on effective social membership" (Wacquant 2012, p. 71). Focusing on the internal logic by which the state acts "as the core agency that sets the rules and fabricates the subjectivities, social relations and collective representations" (Wacquant 2012, p. 66) of membership is useful when exploring how immigration, citizenship, and border regimes are crafted to further the goals of the neoliberal worldview and the supremacy of 'market forces.'

The ideology of market expansion has become naturalized with respect to all human activity, including mobility. Reflecting upon how the notion of 'market forces' has come ideologically dominant in our thinking, Stuart Hall maintains

> 'Market forces' was a brilliant linguistic substitute for 'the capitalist system,' because it erased so much, and, since we all use the market every day, it suggests that we all somehow already have a vested interest in conceding everything to it. It conscripted us ... constantly associating 'the market' with things like freedom, choice—and thus the necessity of a privatised economy; that's the logic. (Hall and Massey 2010, p. 59)

Under the international logic of 'market forces' (that is, the neoliberal capitalist system), the state, then, assumes a role as the core agency that "redraws the boundaries and tenor of citizenship through its market conforming policies" (Wacquant 2012, p. 71). The state does not deregulate; it actively re-regulates the economy in favor of corporations (Vogel 1996). Jones and Novak (1999) referred to this as "retooling the state" in ways that best conform to capital's current interests. Rather than retreating, the state is afforded a key role as the primary "gatekeeper of entrance to and membership of the state" (Robertson and Runganaikaloo 2013, p. 212). As explained above, spatial and spatio-temporal fixes are largely carried out by nation-states through their immigration, citizenship, and border policing regimes. To further neoliberal capitalist expansion, more affluent nation-states are increasingly contracting privatized corporations to run for-profit immigration detention centers and courting poorer nation-states to enter into 'regional agreements' (Golash-Boza 2009; Pearson 2019; Pickering and Weber 2014). These border policing practices effectively privatize human mobility and push the border beyond the territorial borders of the nation state.

As nation-states "politically and legislatively constructed 'the non-citizen' as a counterpart to 'the citizen'" (Wicker 2010, p. 225), they have assumed "deportation power" (Anderson et al.

2011, p. 548) through a range of practices aimed at the expulsion (or potential expulsion) of 'non-citizens' (Anderson et al. 2011; De Genova 2007; De Genova and Peutz 2010; Nail 2015, 2016). Modern nation-states, particularly in the West, have taken what has been called a "deportation turn" (Gibney 2008) in their dealings with unwanted 'non-citizens.' Deportation and detainment powers are key features to understand how the political organization of citizenship and migrant statuses relate to human mobility. Modern nation-states' legal authority to deport and detain without trial only applies to people with 'non-citizen' status. 'Deportability' (De Genova 2002) and 'detainability' (De Genova 2007) denote the ever-present possibility of deportation and "selectively targeted *indefinite* and *protracted* . . . susceptibility for detention" (De Genova 2007, p. 434), which shape the lives of all 'non-citizens,' whether or not they ever experience being deported or detained. Being perpetually subjected to the possibility of deportation and detention makes 'non-citizens' into a group of people who are "deemed to belong to suspect social categories" (Shamir 2005, p. 197). In reference to deportability, "freedom from deportation power—the right to *genuinely* permanent residence—can be seen as one of the few remaining privileges which separates citizens from settled non-citizens in contemporary liberal states" (Anderson et al. 2011, p. 548).

To "understand the pervasive modern division between the citizen and the foreigner" (Hindess 2000, p. 1491), one has to realize how citizenship acts as a "marker of identification" (Hindess 2000, p. 1487). While most discussions of citizenship take place from an 'internalist view,' which is based on Marshall (1950) classic account of "the civil, political, and social aspects of citizenship in the societies of the modern West" (Hindess 2000, p. 1486), it is perhaps more revealing to explore modern citizenship's external, or international, dimensions. When viewed from a broader, externalist perspective, citizenship is best conceptualized as a "dividing practice" (Dean 1999, p. 133) or a technique of "governing a global population of thousands of millions by dividing it into the small subpopulations of particular states" (Hindess 2000, p. 1487). On a global scale modern-day citizenship functions as both a marker of status and a powerful tool for nation-states to enact—and enforce—an "international regime of population management" (Hindess 2000, p. 1496).

Over time, nation-states have appropriated "a certain 'indefinite' power to suspend the law and to fabricate the law" (Butler 2006, p. xvi). As the international system of nation-states construct and normalize a "state of emergency" (Agamben 1998), they create laws that criminalize those who cannot obtain and/or maintain permission to move into and remain within the territorial nation-state. While immigration detention represents the most extreme end of sub-citizenship for people who are construed as 'unlawful non-citizens,' the never-ending 'War on Terror' can be viewed as a normalized and persistent "state of emergency" (Agamben 1998), which justifies the indefinite containment, or detention, of anyone deemed to be a potential threat (Agamben 1998; Butler 2006; Turner 2007).

## 3. Processes of Sub-Citizenship

Current understandings about and practices associated with citizenship and migrant status construct multiple and compounded barriers for certain people to access mobility, human rights, and social justice. Deciding who belongs; who does not belong; who is tolerated, but only for specific purposes and/or periods of time; and who is 'deserving' of citizenship entitlements is carried out through the ways in which powerful social actors construct nationalist identities and nation-states legislate rules governing human mobility and citizenship.

I draw upon the notion of 'sub-citizenship' from recent literature on 'biological sub-citizenship,' which is a new and largely unexplored concept that refers to how neoliberal processes result in people unevenly embodying ill health (Sparke 2016). While Sparke's notion of biological sub-citizenship focused on the health outcomes of people most adversely affected by austerity, the notion of sub-citizenship itself offers a useful way "to elucidate power relations and processes of subordination that simple binary accounts of citizenship" (Sparke 2016, p. 1) tend to obscure. A sub-citizenship lens provides an open-ended, relational approach for exploring how it is that certain people come to

experience different kinds of human rights abuses, dehumanization, and exploitative treatment that is largely, but not entirely, based upon citizenship and migrant status.

As alluded to in the introduction, processes of sub-citizenship apply to everyone, but tend to be more pronounced among those with precarious (and especially illegalized) citizenship and migrant statuses. Experiences of sub-citizenship are not limited to people who move and resettle across international borders under various degrees of choice. Rather, sub-citizenship coordinates the experiences of those with insider ('citizen') and outsider ('non-citizen') statuses. However, those with illegalized migrant and citizenship statuses are particularly vulnerable to most extreme forms of sub-citizenship.

In this article, I explore how processes of sub-citizenship impact upon people positioned near the bottom of the sub-citizenship hierarchy. I focus on children who are categorized as 'illegal migrants' because of what it reveals about the experiences of people positioned at the lower levels of the sub-citizen hierarchy.

I define sub-citizenship as translocal processes of subordination that create various hierarchal conditions of precarity and dehumanization for different groups of people primarily based upon, but not wholly determined by, migration and citizenship status. The sub-citizen thesis maintains that all people are coordinated by powerful, large-scale forces into having certain experiences accessing social justice and human rights due to their historical, social, and legal positioning within transnational regimes of social motion[6], which shape immigration, citizenship, and bordering policies and practices within individual nation-states.

Sub-citizenship is structurally linked to global capitalist expansion and carried out through the international system of nation-states. With "the unceasing drive of capital accumulation" (Heyman 2017, p. 47), nation-states play a key role in determining insiders and outsiders. Nation-states assume the role of the core agency that sets the rules and determines the conditions under which certain groups of people can and cannot access various rights and protections, such as working rights, welfare entitlements, residency rights, and political representation.

Sub-citizenship is produced primarily by nation-states as they enact spatial temporal fixes (Jessop 2006) in response to the pressures asserted by neoliberal globalization. This is accomplished by nation-states' engaging in the following activities: manufacturing citizenship and migrant statuses; reengineering the pathways for people to access mobility; setting the rules to become rights-bearing members of society (i.e., 'citizens'); determining what behaviors are necessary for people to gain and maintain citizenship and other lesser statuses; and containing (and often extracting profits from) populations deemed undesirable.[7]

Sub-citizenship is a new theoretical tool for understanding the ordering of populations along contemporary, neoliberal lines. Applying a sub-citizenship lens offers a critical way to explore how the nation-state manufactures and reengineers citizenship and migrant statuses to further the goals of market expansion. Sub-citizenship may be more appropriate than traditional citizenship or migration perspectives for critically exploring people's experiences with various forms of violence and expulsion, and what role neoliberal globalization has in producing these outcomes.

To understand processes of sub-citizenship we must thoroughly analyze and explore the role that nation-states play in organizing society. In the 1990s, some globalization theorists predicted the decline of the nation-state. However, in recent years, it has become clear that nation-states have assumed

---

[6] Following Thomas Nail's political theory of movement, Kinopolitics, regimes of social motion are metastable social flows that cannot be mapped out in their entirety because they are constantly in motion. As society is always moving within regimes of social motion, borders are not fixed, spatial or even temporal entities. Rather, borders act to "introduce a division or bifurcation of some sort in the world" (Nail 2016, p. 2). Bordering practices direct people to move through regimes of social motion in particular ways.

[7] It is worth noting that among 'citizens' and 'non-citizens' with legal status containment often takes the form of the penal system, while in the case of 'non-citizens' with illegal migrant status containment often takes the form of detention and deportation.

a powerful role particularly with respect to how people can or cannot access mobility, power, and resources. The nation-state's "the modern nation-state is founded on the claim of a homogenous set of citizens whose duty is to protect their common welfare; this necessarily requires the exclusion of others despite liberalism's claims of universalism and equality" (Castañeda 2019, p. 33). To manufacture the appearance of a homogenous set of citizens, nation-states coordinate various forms of social, political, juridical, and territorial expulsion among people who are construed as 'illegal migrants.' Current trends in territorial expulsion are characterized by deportation and arbitrary, prolonged, and, in some instances, indefinite detainment for people with illegalized citizenship and migrant status.

Papadopoulos and Tsianos 2013 describe citizenship as the "toll of sovereign governance that regulates the balance between rights and representation and renders certain populations as legitimate bearers of rights while other populations are marked as non-existent" (p. 182). While those who have citizenship status in their country of residence are legally recognized as rights-bearing members, 'citizens' are not immune from process of sub-citizenship. It is important to understand that those who have (or had) more secure levels of migrant and citizenship status may also experience certain levels of sub-citizenship.[8]

Processes of sub-citizenship coordinate the everyday worlds of people according to where they are hierarchically positioned and their access to resources and protection from various forms of expulsion, such as deportation and detention. Applying a sub-citizen lens helps to systematically unpack the material conditions that structure and shape people's experiences accessing mobility, political representation, and other human rights that tend to be associated with citizenship protections. Through this approach we can see that different degrees of subordination experienced by people are not random, chaotic, or accidental. Rather these conditions are deliberately state crafted through the allocation of citizen and migrant status, which allows or denies access to resources, power, and security.[9]

Figure 1 below is an illustration of how sub-citizenship hierarchically organizes people into different legal statuses. It was specifically designed with the current citizenship and immigration categorizations and rules in the Australian context. Applying Figure 1 to the US context only requires two linguistic substitutions to accurately portray the current citizenship and immigration statuses in the United States. The first is replacing 'non-citizen' with 'alien.' The second is replacing 'unlawful non-citizen' with either 'unauthorized alien' or 'illegal alien.'[10] However, as Australia and most countries in the world use the terms 'citizen'/'non-citizen', Figure 1 uses these terms to illustrate the sliding scales of subordination primarily based upon citizenship and migrant status. Processes of sub-citizenship are not about absolute citizenship and migration categorizations. Rather, the figure directs attention to the unequal relations and social boundaries that are created between groups based on statecrafting citizenship and immigration statuses.

---

[8]   For example, in some nation-states, politicians are deliberating on or enacting legislation that can make it easier to strip citizenship. Creating legislation to strip citizenship status is a technology that extends processes sub-citizenship among those with more secure levels of 'citizen' status.

[9]   This is not to imply that all 'citizens' and 'non-citizens' are treated the same nor do they have the same access to resources, power, and security. This paper is concerned with exploring how citizenship and migrant statuses act as a series of gates that categorically allow or deny access formal rights and protections. Those with citizenship status have more security in terms of residency and other rights, while those with 'non-citizen' status are more vulnerable to all forms of social expulsion, especially territorial expulsion.

[10]  The Immigration and Nationality Act of the United States provides no consistent or overarching definition of the term 'illegal alien,' although the term is used in several provisions under title 8. Conversely, several provisions use the term 'unauthorized alien.'

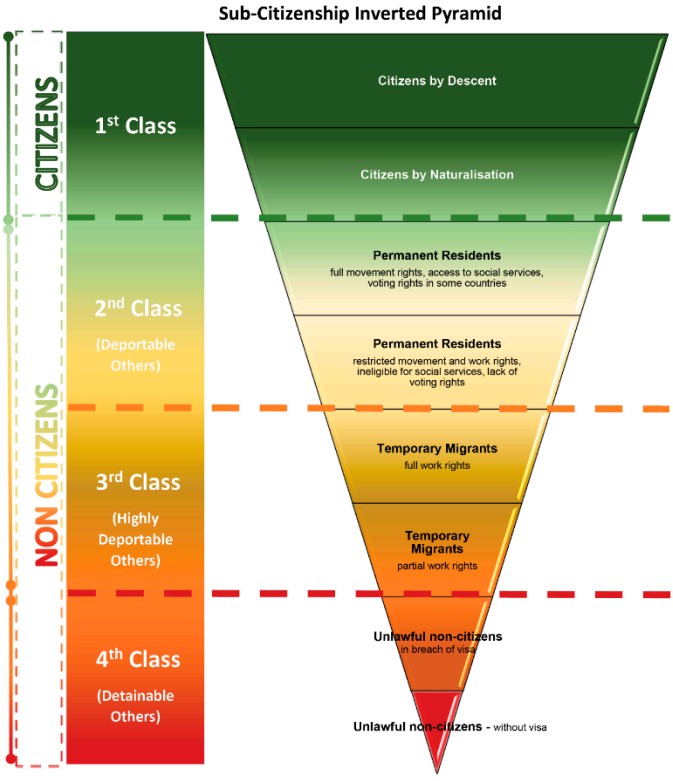

**Figure 1.** Representation of Sub-citizenship hierarchy of statuses.

Figure 1 shows that all people, 'citizens' and 'non-citizens' alike, are hierarchically positioned within processes of sub-citizenship. This figure emphasizes the legal ordering of people that places them into different categories with different degrees of precarity. The nation-state's power to craft legal status serves as the justification to treat people differently depending on their citizenship category. Categorizing hierarchies of people makes it possible for nation-states to enact various forms of exclusion and violence, which are often defended as being necessary to achieve border protection and economic growth. For instance, those who are categorized 'unlawful non-citizens' or 'temporary migrants' face various forms of economic exclusion, such as having no (or limited) legal work options, and social exclusion, as these classes of people are often are ineligible for public services, such as education and health care.

On-the-ground experiences with sub-citizenship can and do vary dramatically between and amongst different groups of people. Processes of exclusion and disadvantage associated with citizenship status can be multiple and compounded for people on the basis of their socio-economic status, race, gender, sexuality, religion, ability, and so on. Those with more secure levels of citizenship status, particularly if they come from dominant social groups, may never think of themselves as being affected by the methods used to determine who belongs to the nation-state as designated by their citizenship status in society. There are many citizens who occupy subordinate positions in society due to poverty, racism, and other forms of discrimination and disadvantage. Certain groups of citizens, such as Indigenous Australians, Native Americans, and black Americans, experience subordinate social, economic, and political statuses regardless of having formal citizenship rights. However, processes of sub-citizenship construct a large social boundary between those with and without citizenship status. In particular, 'citizens' are more secure from the state's use of some kinds of containment technologies, including deportation and immigration detention.

Attaining the legal status of citizenship is often a non-linear and contested journey due to the complex kinds of infrastructure that increasingly produce a 'two-step' or 'staggered' migration process (Hawthorne 2010, Mares 2016) characterized by "non-linear trajectories across multiple statuses and

protracted periods of temporariness" (Robertson 2015). With respect to 'acts of citizenship' and feelings of belonging "citizenship should not be viewed as a linear, but a circular and contested journey" (Mansouri and Mikola 2014). People can move up and down the citizenship ladder because the gradations of sub-citizenship are not stationary, or fixed. Rather, sub-citizenship is a dynamic and fluid process that functions as "sliding scales of subordination" (Sparke 2016, p. 10). The rules for moving up or down the sub-citizenship scale are set by the nation-state and people with different statuses have do display certain desirable characteristics and certain things to maintain or acquire more secure levels of sub-citizenship. For example, children born in Australia whose parents do not have citizenship or permanent residency status are legally classified as 'non-citizens' in their country of birth. If a family living in Australia with 'temporary migrant' status wants to acquire a more secure migrant status (permanent residency) for themselves and their child, one of the things required by the nation-state is for the family members to prove they are healthy. This requires a 'health test' by immigration certified doctors.[11] If the child is found to have a medical condition or a disability, such as autism, it is likely that the family's application for permanent residency would be denied (Soldatic et al. 2012). At this point, the family would slide down the sub-citizenship ladder. If the family does not 'voluntarily' leave the country, then they would be territorially expelled by the nation-state using the technologies of detention and/or deportation.

Referring to Figure 1, those who enter the country with a permanent visa (that is, through the humanitarian or the migration program) are positioned in Australia and America as 'second class non-citizens.' People who hold permanent visas are the most secure group of 'non-citizens.' This group of people have obtained a certain degree of protection from territorial expulsion because they have the nation-state's permission to stay indefinitely under certain conditions. They also have a certain degree of protection from economic expulsion, as they have been authorized with unrestricted work rights by the nation-state. Although in some cases, permanent residents must work in their nominated skilled occupation and in specific geographic locations. Oftentimes, this group experiences expulsion from social welfare provisions as well as political expulsion, as they cannot vote or run for office. This group may go on to become 'first class citizens' by naturalization, but the rules and processes determining the pathways toward citizenship in many countries are increasingly becoming longer, more difficult, expensive, and contingent on behavior (Anderson et al. 2011; Hugo 2014; Mares 2016).

Non-citizens who are construed as 'unlawful' (often called 'illegal migrants') become detainable and deportable Others. It is possible for people to be detained for short periods of time and then either be deported or allowed to stay. However, I position detainability at the bottom of the sub-citizenship ladder, as being detained—especially indefinitely detained—puts people 'outside the pale of the law' (Arendt 1951). Without the protections of the law, detainable Others are increasingly torturable Others. Detainment without the right to a trial or appeal is the most precarious, uncertain, and arguably, the most dangerous form of territorial expulsion there is.

While citizenship delivers certain benefits to those who are citizens of the state in which they live, it "also has a variety of other, less obviously benign consequences" (Hindess 2000, p. 1495). Here, Hindess alludes to an externalist view of citizenship where citizenship status acts as a marker of difference that enables the nations-state system to manage the flow of international populations. However, when viewing citizenship internally (that is, within the nation-state), we can also clearly see that having citizenship status is not the panacea for people to access human rights and social justice, as many other factors including racial or ethnic discrimination, poverty, and the particular economic and human security conditions of the nation-state in question place limits on the advantages of having citizenship status. Along with citizenship and migrant status, other factors that influence individual experiences

---

[11]　The Health Requirement began with the inception of Australia's *Immigration Restriction Act* (1901). Since 1901, potential all migrants to Australia, including those with refugee status who come through the Humanitarian Program, are required to undergo a cost-benefit assessment of health. Under this neoliberal approach to migration, people who have a disease or disability are actively excluded from the Australian migration process.

with sub-citizenship include historical time, physical location, and the individual circumstances of people. It should be noted that those who are highly-skilled and/or have access to support, in particular financial assets and capital, have a clear advantage in accessing mobility and human rights in a global capitalist system that has increasingly commodified mobility and reengineered immigration and citizenship pathways to expand economies.

As capitalist expansion is structurally compelled to place controls on the mobility of labor (Mezzadra and Neilson 2017; Nail 2015, 2016; Neilson and Rossiter 2005; Sheller 2011), processes of sub-citizenship function on a global scale. Most of the world's population are structurally bound to experience different degrees of (im)mobility, dehumanization, extraction, and exploitation in response to and to further the goals of capital accumulation and expansion[12]. People who live in countries where they have citizenship status are usually afforded certain rights and protections from the state's power to deport and detain. Those who live in countries with more privileged migrant statuses, such as 'permanent residents,' also have certain rights and protections, although to a far lesser degree than 'citizens.' However, as "those who are excluded from the human rights framework are the same persons who are excluded from the citizenship-rights framework" (Sagy 2013, p. 231), people who live in counties where they are not formal members (i.e., 'citizens') are more likely to experience precarity and lack of access to human rights. 'Non-citizens' are considerably more at risk of experiencing various forms of (im)mobility, dehumanization, extraction, and exploitation because they occupy subordinate social and legal positions. People who live in a nation-state and are not considered 'citizens' of that state face compounded barriers to accessing human rights and social justice, and are largely unprotected from arbitrary state-led violence. This is particularly the case when one's mobility across or presence within the nation-state is interpellated as being 'illegal' by the current immigration, citizenship, and border regimes of the nation-state.

In the next section, I apply a sub-citizen lens to explore the experiences of children with 'illegal migrant' status in the USA and Australia. To illustrate processes of sub-citizenship among those are lower levels of the sub-citizen hierarchy, I draw upon the immigration detention policies in Australia and the United States because they are arguably among the most extreme examples in the contemporary world. Drawing upon the words and pictures of children in detention, I link their experiences with sub-citizenship to the "territorial solutions" (Cornelisse 2010) used to fix the 'problem' of people who attempt to move without the nation-state's authorization. The experiences of detained children are drawn upon to explore how processes of sub-citizenship are connected to violence. Understanding how people at the lower levels of the sub-citizen hierarchy are affected by the nation-state's deportation and detention powers is particularly relevant and timely, as many countries around the world are increasing their powers to banish, exile, and torture people who are perceived as risky, threatening, or needy.

## 4. Children in Immigration Detention

The United Nations High Commissioner for Refugees estimates that 50 million children had migrated across international borders or were forcibly displaced in 2016. Twenty-eight million (1 in every 80) children migrated due to conditions of violence and insecurity (United Nations International Children's Emergency Fund (UNICEF) 2016). In many countries, such as the United States, immigration detention statistics are not readily available, resulting in a widespread lack of transparency that makes genuinely informed public and policy debate nearly impossible. Globally, there is no validated data detailing the number of children held in immigration detention; however, the number is estimated to be in the millions (Inter-Agency Working Group (IAWG) to End Child Immigration Detention 2016).

---

[12]　While all people are positioned somewhere on the citizenship's spectrum, economics elites, certainly those of the billionaire class, cannot be said to experience human (im)mobility, dehumanization, extraction, and exploitation.

Beginning the 1980s, all major Western states practice what they call 'civil' or 'administrative' immigration detention (Skodo 2017). This means that nation-states classify immigration detention as an administrative policy, rather than a punitive one. However, despite being characterized as civil confinement, a closer look at the immigration detention regimes around the world reveals that this form of forced enclosure and confinement carried out against undocumented migrants is often many degrees harsher than criminal detention. In addition, people facing immigration detention lack access to the same constitutional protections as those facing criminal incarceration (Groves 2004). Australia and the United States also follow this trend of considering their immigration detention policies as being 'administrative.' For example, the Australian Border Forces website states: "In Australia, immigration detention *is administrative* not punitive. It helps us manage our temporary entry and permanent migration programs" (Australian Border Force (ABF) 2019). This statement issued by the Australian government also illustrates the textual mediation of the social relations coordinating the nation-state's immigration detention regime.

The practice of mandatory detention of people with 'unlawful non-citizen'[13] status is a relatively new invention of neoliberal statecraft in both countries. Mandatory detention began in the United States in 1988 under the Reagan government and in Australia in 1992 under the Keating government. Currently, both countries practice mandatory detention to hold adults and children in immigration detention centers[14] if they are construed as being 'illegal migrants.' While it is beyond the scope of the present paper to review in detail the similarities and differences between the two countries' approaches to mandatory detention, it should be noted that with regards to children in detention:

(1) The US detains far more children than Australia.
(2) The US lacks transparency regarding the number of children in detention.
(3) The US is the only country in the world that systematically separates and detains immigrant families en masse (Wood 2018). Australia is the only country in the world with a policy of mandatory detention and offshore processing of people seeking asylum who arrive without a valid visa (Australian Human Rights Commission (AHRC) n.d.).[15]
(4) The average length of time children are held in immigration detention is believed to be roughly similar: 3.5–7.9 months in the USA and 7.5 months in Australia (Australian Human Rights Commission (AHRC) 2014).
(5) Both countries have been condemned by the United Nations and other human rights organizations for breaching international human rights law, including the Rights of the Child (Australian Human Rights Commission (AHRC) 2014; Amnesty International 2013, 2015, 2015; Nethery and Holman 2016; United Nations News 2018).

Regarding publicly available data on the numbers, locations, and durations of children held in immigration detention centers, the United States is far less transparent than Australia. It is not known how many children are held in US detention centers because the Office of Homeland Security does not issue these figures. It is estimated that there were 15,000 children detained in over 100 immigration detention centers in the USA in 2018. In Australia in 2014 there were 1068 children held in immigration

---

[13] 'Unlawful non-citizen' designates a legal status. People with this status are often referred to as 'illegal migrants' in popular discourses, including politics and media.

[14] In this article, I use the term 'immigration detention center' to ensure consistency when discussing immigration detention in Australia and the United States. However, the term most often used by the United States government is 'Customs and Border Protection facility' (CBP), while the Australian government's preferred term is 'offshore processing center.' Additionally, in the interest of consistency, this article uses the American spelling of 'center' for both US and Australian immigration centers, noting that the Australian spelling of the word is 'centre.'

[15] In July 2019, US President Trump enacted the "third party asylum rule". Under this rule the US government plans to ban asylum seekers form making a claim and send them to offshore detention in another country, such as Guatemala or Mexico (Pearson 2019). This approach was first trialed in Australia and referred to as 'offshore detention.' It extends border enforcement beyond the nation-state's territorial border and into the borders of other neighboring nation-states.

detention centers in three locations: mainland Australia (584 children), Christmas Island (305 children), and Nauru (179 children) (Australian Human Rights Commission (AHRC) 2014).

Over the past several years, fewer children are being held in Australian detention centers. By 2018, less than 10 children were being held in Australian offshore and onshore immigration detention centers (Department of Home Affairs 2018). In February 2019, the Department of Home Affairs announced that the final four children on Nauru were flown to the US for resettlement with their families.[16] Several hundred people were resettled in the US as part of the US refugee deal. In August 2019, Australia reopened the Christmas Island Detention Center to detain one Tamil asylum seeker family of four from Sri Lanka whose two young daughters were born in Australia. This case has resulted in a large public outcry to return the family to their home in Biloela, Queensland (Doherty 2019).

There are numerous reasons why there are currently less children being held in offshore Australian detention centers. No asylum seeker boats have reached Australia since 2013, when the 'Operation Sovereign Borders' began the practice of naval turn backs. This practice means that when people traveling by boat do not have authorization (visas) to travel to Australia, their boat is turned back. The awareness raising work of the Australian Human Rights Commission, Amnesty International, researchers, journalists, and advocates have influenced public opinion and pressured the government to get children out of offshore detention. However, in most cases, asylum seekers and refugees have been shuffled into other forms of detention, such as community detention. Others are living in Australia with visas that assign them highly precarious, 'non-citizen' statuses with little access to welfare benefits and no pathways available to gain citizenship rights and protections.[17]

The increased use of immigration detention in many countries around the world has raised significant human rights concerns in recent years. In 2019, the UN Human Rights Chief stated that she was "deeply shocked that children are forced to sleep on the floor in overcrowded facilities without access to adequate healthcare or food, and with poor sanitation conditions" (United Nations News 2019). The UN Chief stated that immigration detention is never in the best interests of a child and urged the authorities to find non-custodial alternatives for migrants and refugee children and adults. The Australian Human Rights Commission has repeatedly warned that Australia's practice of mandatory has led to numerous breaches of human rights, particularly among children. The Commission found that mandatory and prolonged immigration detention of children is in clear violation of the Convention on the Rights of the Child (Australian Human Rights Commission (AHRC) 2014). Amnesty International has issued numerous reports concluding that the United States and Australia's immigration detention systems are in breach of international human rights obligations (2011, 2013, 2015, 2016).

Although there is overwhelming evidence that holding people in immigration detention is physically and mentally harmful to adults and children (Australian Human Rights Commission (AHRC) 2014; Amnesty International 2011, 2013, 2015, 2016; Inter-Agency Working Group (IAWG) to End Child Immigration Detention 2016; Keller et al. 2017; Mares et al. 2002; Moss 2015; Wood 2018), the US and Australia are among over 100 countries worldwide that are known to detain children for migration-related reasons (Wood 2018).

The following sections draw upon major reports and media coverage detailing the experiences of children held in immigration detention centers in Australia under the 'Operation Sovereign Borders' policy and the United States under the 'Zero Tolerance' policy.

---

[16] In 2017, the US agreed to consider resettling refugees held in Australia's offshore detention centers on Nauru and Manus Island, as well as those who have been transferred back to Australia for medical reasons. Often referred as the 'US refugee deal,' this arrangement has resulted in 619 refugees being resettled in the US (Kaldor Centre for International Refugee Law 2019).

[17] There are approximately 30,000 asylum seekers living in Australia who were formally held in offshore detention and have been granted visas allowing them to live in Australia without access to pathways citizenship. This group have become known in public policy as the 'legacy caseload.' Most of these people were held in offshore detention since 2013.

*4.1. Experiences of Children Held in Australia's Immigration Detention Centers*

In 2014, the Australian Human Rights Commission (AHRC) released a report entitled *The Forgotten Children: National Inquiry into Children in Immigration Detention.* The report concluded that "immigration detention is a dangerous place for children" (Australian Human Rights Commission (AHRC) 2014, p. 10). Drawing upon the stories of people in detention; professional evaluations of psychologists, doctors, and teachers working with them; and artefacts, such as pictures drawn by the children and adults held in detention, the report conveys the dangerous environment of detention centers and provides compelling first-hand evidence of the impact prolonged immigration detention has on the mental and physical health of children and adults.

During a 15-month period, from January 2013 to March 2014, the AHRC collected the following statistics of violence occurring at Australian detention centers:

- 57 serious assaults.
- 233 assaults involving children.
- 207 incidents of actual self-harm.
- 436 incidents of threatened self-harm.
- 33 incidents of reported sexual assault (the majority involving children).
- 183 incidents of voluntary starvation/hunger strikes (with a further 27 involving children) (Australian Human Rights Commission (AHRC) 2014, p. 62).

These statists indicate that children and adults held in Australia's immigration detention centers are subjected to high rates of violence and abuse in detention centers.

Overwhelmingly, the children interviewed by the AHRC described their experiences living in immigration detention in a highly negative light. The children, some as young as pre-school age, used terms including 'jail,' prison-like,' 'depressing,' 'die,' 'no hope,' and 'crazy-making' to describe immigration detention. Below are examples of statements made by children held in immigration detention interviewed by the AHRC (2014):

My hope is finished now. I don't have any hope. I feel I will die in detention.

17-year-old child, gender redacted, held at Phosphate Hill
Detention Center on Christmas Island

My country and my religion is a target for Taliban. There were bomb blasts and always big wars and terrible attacks. Shia people have arms, legs, noses hacked off, necks slashed, plus there is rocket fire and missiles. This is because I am Shia. All this means no one is safe and how because I escaped. I am in detention.

Child, age and gender redacted, held at Nauru Regional
Processing Center

I feel like I'm in jail, no one here to help us. It's just me and God.

17-year-old child, gender redacted, held at Christmas
Island detention Center

These statements capture the severity of hopelessness, isolation, depression, and ongoing trauma experienced by children held in Australian immigration detention centers. As people who sought asylum, many of the children had direct experiences with trauma as they and their families fled conditions of war and conflict. In other cases, children explained that their families attempted to move to Australia to get away from oppressive regimes. One child held at Nauru Regional Processing Center articulated (Australian Human Rights Commission (AHRC) 2014):

> I am a thirteen years old boy that came to Australia with my parents and my eight-year-old brother for better and brighter future. We took the risk of this dangerous way because we had no other option. I heard Australian politicians say Iranian people come to Australia because of their economic problems. But we weren't poor in my country. We weren't hungry, homeless, jobless, and illiterate. We immigrate because we had no freedom, no free speech, and we had [a] dictatorship.

Whether the children and families moved to escape from war, conflict, dictatorships, economic and safety insecurities, or, in many cases, a combination of factors, the children describe ongoing and compounded trauma from their experiences of being held against their will in Australian detention centers.

The Australian government responded to the allegations of abuse in the AHRC's report by conducting an internal investigation called the Moss Review. Confirming the AHRC's findings, the Moss Review found that there were both reported and unreported allegations of sexual and other physical assaults inflicted on children in detention. The report calculated that between October 2013 and October 2014, 17 minors in detention engaged in self-harm (including one attempted hanging). The youngest child involved in self-harm was 11 years old (Moss 2015).

In 2016, Amnesty International released a report entitled *Island of Despair: Australia's "Processing" of Refugees on Nauru*. The report provides detailed evidence from hundreds of interviews among people in detention, their family members and service providers on Nauru showing that people held in detention are subjected to a "regime of abuse and neglect (that) does not even spare children" (p. 18). The report strongly condemns Australia's current immigration policies and its practice of mandatory detention, stating:

> the Government of Australia has made a calculation in which intolerable cruelty and the destruction of the physical and mental integrity of hundreds of children, men and women, have been chosen as a tool of government policy. In doing so the Government of Australia is in breach of international human rights law and international refugee law. The conditions on Nauru—refugees' severe mental anguish, the intentional nature of the system, and the fact that the goal of offshore processing is to intimidate or coerce people to achieve a specific outcome—*amounts to torture*. (p. 7, my emphasis)

These reports establish a strong link between the mental and physical deterioration of people "trapped in limbo" facing "debilitating uncertainty about their future" (p. 22) while living in conditions of indefinite immigration detention. Children held in immigration detention were found to have significantly higher rates of mental disorders than other children in Australia (AHRC n.d.). A pediatric and child health expert in assisting the national inquiry into children in immigration detention stated:

> I was particularly distressed by the utter despair of the unaccompanied boys I spoke with on Christmas Island—despair underpinned by past, present and anticipatory trauma. Young men, in the prime of their lives, who face the intolerable realization that any hope of a better life had almost evaporated. (Australian Human Rights Commission (AHRC) 2014, p. 150)

Since Operation Sovereign Borders commenced in 2013, health professionals, including child psychologists, have repeatedly raised concerns regarding the physical and mental effect that prolonged detention was having upon children. Media has reported that young children held in detention were expressing the desire to no longer live. Some stopped eating, drinking, and talking. Others were self-harming. In 2018, a 12-year-old boy, who had been held in immigration detention in Nauru since he was 8 years-old, was evacuated after refusing to eat for 20 days. The Australian Border Force repeatedly tried to block his medical evacuation (Doherty 2018).

Figures 2–4 below are reprinted from the Australian Human Rights Commission's *Forgotten Children* Report. These drawings provide another window into the extreme levels of psychological

stress experienced by preschool, primary, and secondary school-aged children held in Australian immigration detention centers (Australian Human Rights Commission (AHRC) 2014).

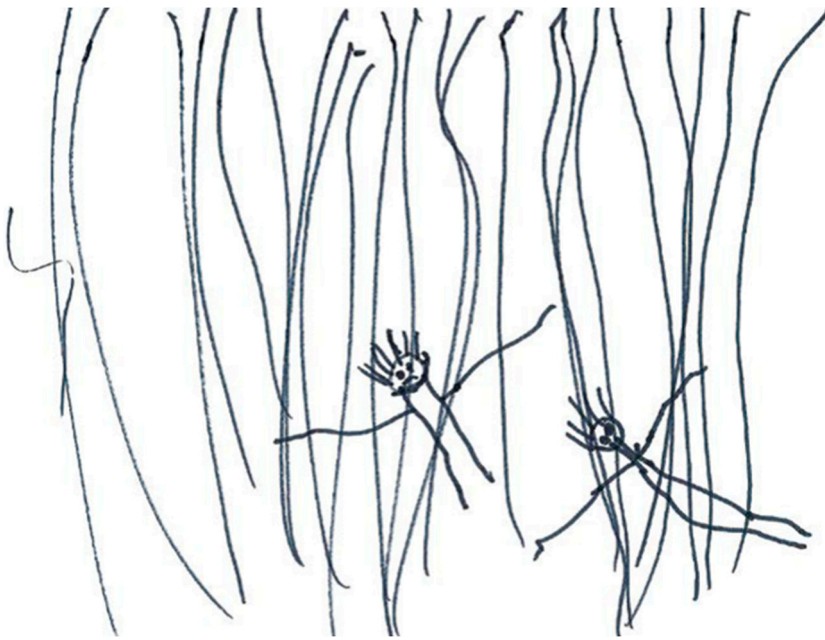

**Figure 2.** Drawing by preschool school-age girl, held at Christmas Island Detention Center. © *Australian Human Rights Commission 2014. The Forgotten Children: National Inquiry into Children in Immigration Detention 2014.*

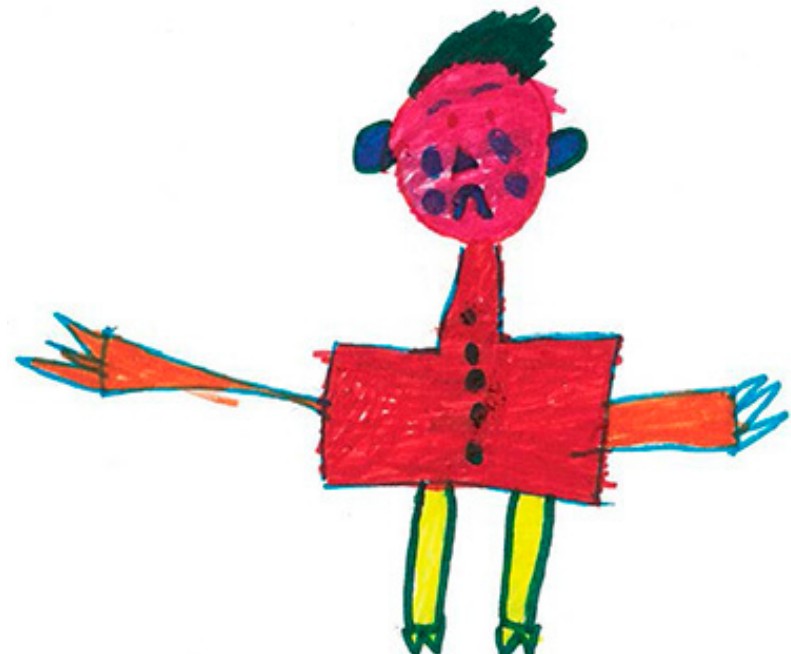

**Figure 3.** Drawing by primary school-age child, gender redacted, held at Christmas Island Detention Center. © *Australian Human Rights Commission 2014. The Forgotten Children: National Inquiry into Children in Immigration Detention 2014.*

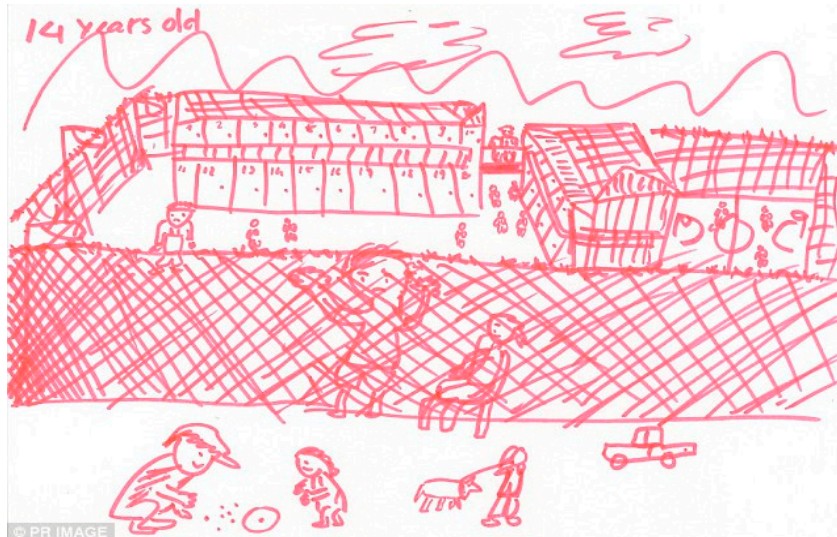

**Figure 4.** Drawing by secondary school-age child, gender redacted, held at Darwin Detention Center. © *Australian Human Rights Commission 2014. The Forgotten Children: National Inquiry into Children in Immigration Detention 2014.*

Detained children as young as pre-school age have communicated clear signs of trauma, depression, and despair by drawing pictures of being trapped in cages and crying. Other children in Australian detention have tried to make sense of what they were experiencing by drawing pictures of blood, weapons, and barbed wire. As will become evident in following section, children held in US detention centers have communicated similar feelings of despair, depression, and trauma though their words and drawings.

### 4.2. Experiences of Children Held in US Immigration Detention Centers

In the United States, there has not been the same level of independent investigation detailing the conditions of children held detention centers as has been done in Australia by the Australian Human Rights Commission, the Moss Review, and Amnesty International. This again reveals that immigration detention and the details surrounding the treatment of children in detention is far less transparent and more secretive in the United States than in Australia.

However, after a flu outbreak that sent five infants held in US immigration detention to hospital, lawyers from the Center for Human Rights and Constitutional Law and a border certified physician were granted access to the Ursula Detention Center in McAllen, Texas, which is the largest immigration detention center in the United States. In June 2019, the team was briefly allowed into the detention center to monitor government compliance with the Flores settlement, a 1997 agreement that requires the government to keep children in 'safe and sanitary' conditions while being held in immigration detention. More than 1000 children were being held at the detention center with the team interviewing and clinically examining as many children as they could. However, the team was not permitted to enter the area where many of the sickest kids were being held in cages (Raff 2019).

The voices of three children held in detention who were interviewed by the Center for Human Rights and Constitutional Law provide a glimpse into the conditions faced by an unknown number of infants, children, and teenagers currently held in US detention centers (McLaughlin 2019).

> I'm in a room with dozens of other boys. Some have been as young as 3 or 4 years old. Some cry. Right now there is a 12-year old-boy who cries a lot. Others try to comfort him. One of the officers makes fun of those who cry.
>
> 　　　　　　　　　　　　　　17-year-old boy, currently held in US immigration
>
> 　　　　　　　　　　　　　　detention center in Clint, Texas

> A Border Patrol agent came into our room with a 2-year-old boy and asked us, "Who wants to take care of this little boy?' Another girl said she would take care of him but lost interest after a few hours and so I started taking care of him ... I feed the boy, change his diaper, and play with him. He is sick. He has a cough and a runny nose and scabs on his lips.

> > 15-year-old child, gender and detention center location
> > redacted

> I have been here without bathing for 21 days. I have seen that when we try to complain about the conditions the (officers) want to know what we said. Then they start yelling at us, saying things like, 'You don't belong here,' 'Go back to where you came from,' 'You are pigs,' 'You came here to ruin my country.' They try to intimidate us. I have seen officers hit other detainees in the stomach.

> > Age redacted, young mother held at the Ursula detention
> > center in McAllen, Texas

The children interviewed described horrific conditions of being separated from their parents and warehoused in detention centers where they are subjected to psychological abuse, mistreatment, and child neglect. More specifically, the children's voices above explain that they have been held in overcrowded rooms for extended periods of time, subjected to verbal harassment, intimidation and physical abuse by detention officers, older children are caring for younger children, and sick children are not being properly treated.

Since the 'Zero-Tolerance' policy began in 2018, at least seven children have died in US custody. Between 2014–2018, the Office of Refugee Resettlement received 4556 complaints, including allegations of sexual abuse or sexual harassment of migrant children. There have been numerous reports of severe overcrowding, unsafe, unsanitary, and inhumane conditions in US detention facilities housing an estimated 15,000 children with 'illegal' status in the United States (Chalabi 2018). According to the Office of Inspector General (2019) report, at least four immigration detention centers were found to pose "immediate risks or egregious violations of detention standards ... including nooses in detainee cells ... and significant food safety issues" (p. 3).

After assessing 39 children under the age of 18 held at the Ursula facility, Lucio Sevier, a board-certified physician stated in an interview:

> the conditions within which they are held could be compared to torture facilities. That is, extreme cold temperatures, lights on 24 h a day, no access to medical care, basic sanitation, water or adequate food ... every single person I spoke to [was] denied access to hand-washing even after bathroom use. (Novack 2019)

Doctor Sevier concluded that the unhygienic conditions in which the children were being held is "tantamount to intentionally causing the spread of disease" (2019). The Center for Human Rights and Constitutional Law report cited that babies were being cared for by other children and that children over 6 months were not provided age-appropriate meal or clean bottles. According to Doctor Sevier, "to deny parents the ability to wash their infant's bottles is unconscionable and could be considered intentional mental and emotional abuse" (Marshall et al. 2019).

The report uncovered that children did not have access to soap, toothbrushes, toothpaste, and many children were forced to sleep in freezing conditions on concrete floors while held in the Ursula detention center for extended lengths of time. While these facilities are only legally permitted to hold children up to 72 h, many of children reported being detained for nearly a month in unsafe and unsanitary conditions that are in breach of the Flores Settlement.

In July 2019, three children who were held in detention centers and separated from their families and relocated to a respite center were asked to draw pictures about their time in detention. Figures 5–7

below are the children's pictures representing their memories and impressions of the time they spent in immigration detention.

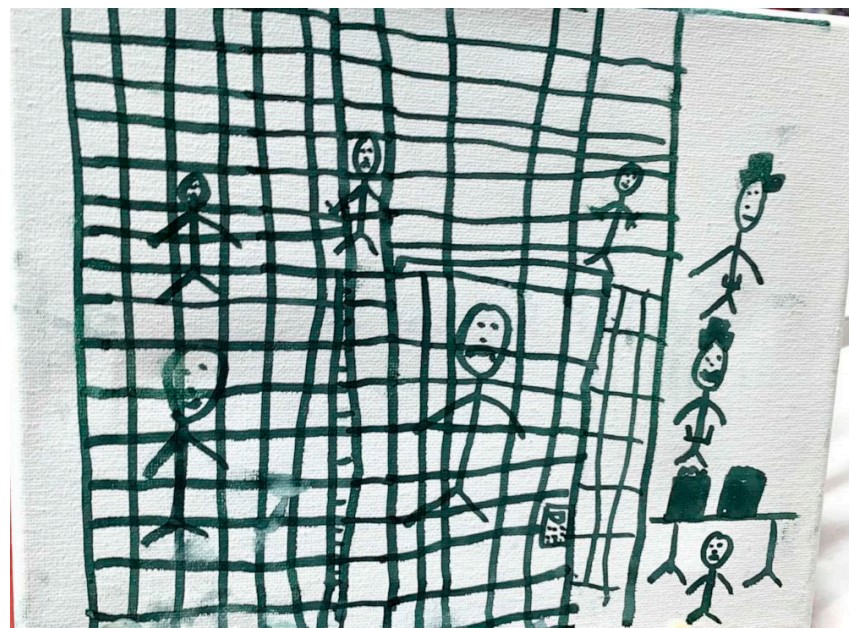

**Figure 5.** Drawing by primary school age boy held in an unspecified US detention Center, 2019. Courtesy of American Academy of Pediatrics.

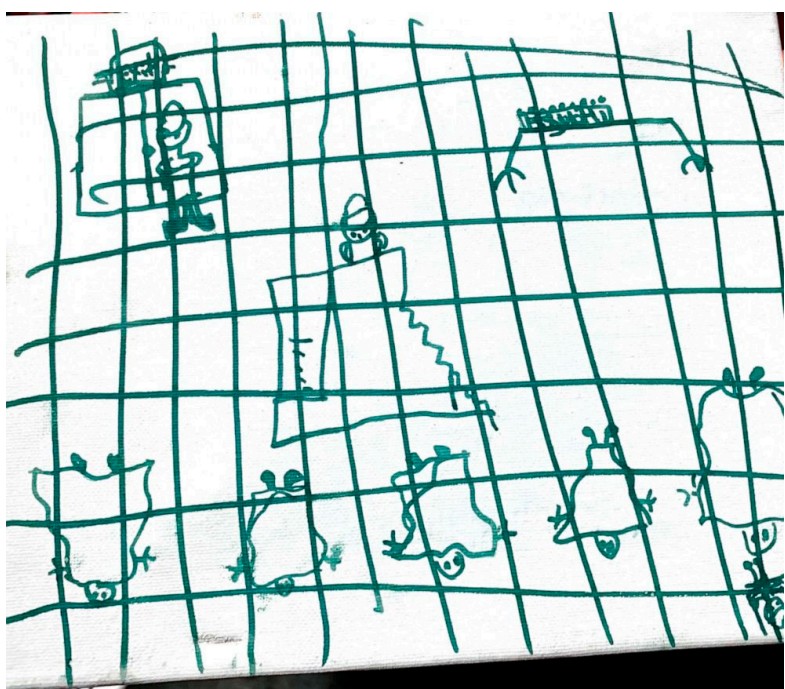

**Figure 6.** Drawing by primary school age child, gender redacted, held in an unspecified US detention Center, 2019. Courtesy of American Academy of Pediatrics.

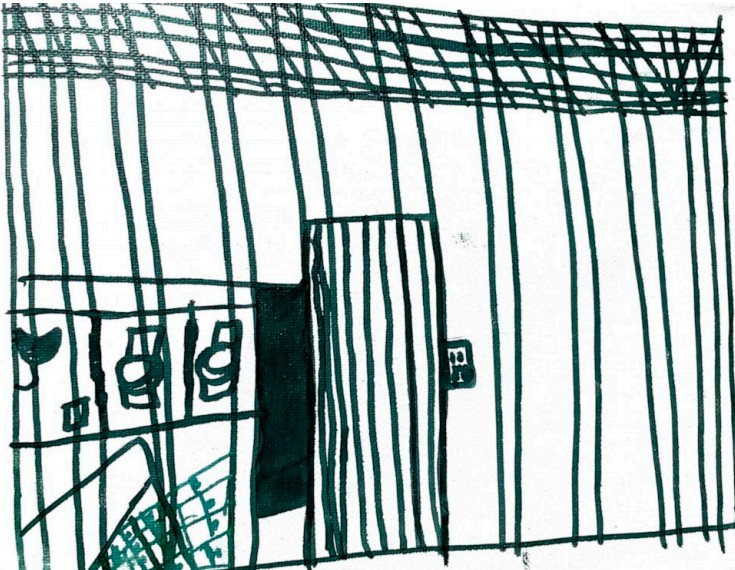

**Figure 7.** Drawing by primary school age child, gender redacted, held in an unspecified US detention Center, 2019. Courtesy of American Academy of Pediatrics.

The pictures in Figures 5–7 were released by the American Academy of Pediatrics (AAP) in July 2019. It is not known which facilities the children who drew the pictures were held in nor how long they spent in detention, however, it is possible they were held in the Ursula center before moving on to the Respite Center. The words and drawings of the children held in U.S. immigration detention centers are strikingly similar to that of the children held in Australian immigration detention centers. In both Australia and the U.S. contexts, children expressed their experiences with trauma, fear, and sadness though their drawings of people trapped in cages, sad faces, and tears. In Figure 6, a primary school age child drew people sleeping in a concentration camp like setting. This drawing poignantly depicts the harsh, overcrowded, and inhumane conditions a child experienced while held in a U.S. immigration detention center.

Having established some of the ways nation-states enact violence upon children construed as 'illegal migrants,' the next section takes a step back to denaturalize the often taken for granted concepts of the nation-state, citizenship, and borders regimes. As nation-states create various categories of human though hierarchies of personhood (citizenship), people's experiences with violence, abuse, torture, and suffering are rendered as merely the collateral damage of border protection.

## 5. Denaturalizing the Nation-State, Citizenship, and Borders

Ideological constructs work "hard to make certain things 'obvious,' and the more we find these things obvious, self-evident and unquestionable, the more successfully the ideology has done its job" (Zupancic 2011). Grand concepts such as the international system of nation-states and supporting concepts such as citizenship, border protection, and 'illegal migration' are often presented as being neutral, empirical 'facts' devoid of any ideology. As 'facts,' nation-states, citizenship, borders, and migrant status structure the limits and set the frameworks within which we think about, discuss, value, and treat people. These constructs yield enormous power because they shape social understandings, public discourses, policies, and practices that determine which groups of people are permitted and welcomed to physically occupy certain spaces and have access to various rights in society. Nation statehood, accompanied by citizenship, and more recently, border policing, have largely become taken-for-granted as obvious and unquestionable necessities for organizing society.

Applying a historical lens to the contemporary system of nation-states and border regimes and problematizing traditional migration and citizenship perspectives allows us to denaturalize the

contemporary nation-state ideology and the notions of citizenship and border policing that sustain this grand narrative.

*5.1. Historizing the Nation-State*

When research leaves uninterrogated the assumption that human mobility ought to be governed by the container-like logic of the international system of nation-states, it produces, perpetuates, and naturalizes a host of unspoken, but implicit, power dynamics. As explained above, traditional migration and citizenship perspectives necessarily frame border-crossing mobile people as risky, deviant figures who lack permanence within the international nation-state system (Bauder 2013; Nail 2015; Turner 2007). However, the historical record clearly shows that humans were geographically mobile long before invention of modern-day nation-states. The social, political and historical conditions coordinating people's experiences with movement created nation-states themselves (Nail 2015, 2016).

When thinking about contemporary issues related to human (im)mobility and border securitization regimes, it is worthwhile to conceptually step back from the present and remember that in historical terms, the international system of nation-states and the notion of citizenship are relatively new concepts. The world-wide nation-state building project started in earnest between World Wars One and Two, making this the globalized form of social and political organization just over one hundred years old. Rather than limiting migration research to the terms and political boundaries determined by nation-states, we must materially explore people's on-the-ground conditions with (im)mobility in the context of the historical conditions giving rise to nation-states (Tilly 2005).

From its conception, modern statehood has been primarily concerned with the construction of 'nations,' a task which it continues to use to justify its own existence (Anderson 1983). As exclusive 'national' societies were being formed, the newly emerged nation-states began to regulate social mobilities of people within their territories, primarily through the creation of policies affecting people's access to education, employment, housing, welfare, health care, and so on. At the same time, nation-states began to restrict the mobilities of people moving across their borders (Raithelhuber et al. 2018). As the international system of nation-states took shape, sovereign, territorially bounded nation-states formed and were solidified through the creation of nationalities and national identities.

Human migration across and through nationalized spaces became regarded as a problem only recently in history (Sharma 2006). The invention of citizenship was a technology that helped to solidify the state's monopoly on the freedom of movement (Torpey 2000). Through the construction of citizenship and immigration regimes, nation-states set the rules by which certain people are construed as formally 'belonging' to or being 'members' of a nation-state. Statecrafting of citizenship and migration legislation was aided by the invention of passports and visas, which were—and continue to be—key technologies used by nation-states to set the rules of belonging and people's access to mobility (Torpey 2000). As states began to spatially regulate people's physical movements across and within international borders, they decreased various forms of security for people who were construed as 'non-citizens' (Raithelhuber et al. 2018). At the same time, some states (particularly the more affluent West) began to nationalize welfarism for those construed as 'citizens.' In recent years advances in surveillance technologies, such as closed-circuit television (CCTV), biometrics, and trusted traveler databases, have enabled nation-states and other mobility gatekeepers, such as airports, to more closely monitor and police human mobility (Adey 2004, 2006; Sheller 2016; Weber 2011).

Since the post-World-War-Two era, the global pattern of distinctive, sovereign nation-states has framed nearly every aspect of social, political, and economic life for people around the world. As we are "dominated by an image of the world in which the most basic patterns of social relationships take place within distinct, sovereign nation states" (Raithelhuber et al. 2018, p. 1), these social constructions shape how we view human mobility and social belonging. Living in a world that has been divided into territorially defined nation-states has great influence upon our conventional understandings of

migration and citizenship, both of which are "firmly entrenched in the concept of the nation-state" (Bauder 2013, p. 56).

Migration and citizenship studies have not been immune from the ideological influence and global dominance of nation statehood. The unquestioning adherence to nationalist thought and institutions is most evident in the state-centered approach taken by mainstream studies on citizenship, migration, forced migration, demography, and human geography. Scholarship that takes the national partitioning and ordering of the world as natural is deeply entrenched in 'methodological nationalism' (Agnew 1994; Bauder 2013; Beck 2007, 2009; Wimmer and Schiller 2002; Amelina et al. 2012). When migration research fails to critique the nation-state scale it falls into an "epistemological trap" (Agnew 1994) of leaving uninterrogated the social, historical and political power dynamics shaping people's experiences with human (im)mobility and belonging.

Although it is "difficult to escape from the categories we inherit" (Isin and Nyers 2014, p. 6), it is incumbent upon researchers to denaturalize, deconstruct, and interrogate the institutional machinery underpinning the international system of nation-states. To disrupt the international system of nation-states and the rise of militarized bordering practices, it is necessary to explicate the links between the intensification of national borders with the development of late, or neoliberal, capitalism (Mezzadra and Neilson 2013, 2017; Stratton 2009; Nail 2015, 2016; Papadopoulos and Tsianos 2013; Shachar 2009).

At this historical juncture characterized by the intensification of border policing regimes and the related human rights violations experienced by increasingly numbers of people deemed as 'undesirable' migrants or as citizens, critical research and analysis is desperately needed to explicate the heightened policing powers of nation-states and interrogate notions of membership and rights allocation based on the shifting rules pertaining to citizenship status. Migration research must depart from one-dimensional, state-centric, institutionalized, and policy-oriented knowledge that produces both the national ordering of the world and the very categories of 'citizen' and 'non-citizen' themselves (Raithelhuber et al. 2018).

## 5.2. Problematizing Traditional Migration and Citizenship Perspectives

It is fundamental to recognize that human mobility is not unique to contemporary times. Research in the new mobilities paradigm "starts from the premise that people have always been on the move, but that human mobilities have been variously valued and interpreted through time and within as well as across cultures and societies" (Salazar 2016, p. 6). By focusing attention on the historical constancy of movement, new mobilities research considers "mobility as a natural tendency in society" (p. 3), thereby naturalizing human movement "as a fact of life and as a general principle that rarely needs further justification" (p. 3).

Bauman (1998), Sheller (2011), and others have noted that the forces of neoliberal globalization have transformed the ways in which people migrate and move. While more people are on the move, "the world is arguably moving differently, and in more dynamic, complex and trackable ways than ever before" (Sheller 2011, p. 1). In this age of migration unprecedented numbers of people are "facing more challenges of forced mobility and uneven mobility" (p.1), whereby access to mobility has become "the most powerful and most coveted stratifying factor" (Bauman 1998, p. 2) among people in the modern world.

Traditionally, migration and citizenship perspectives are applied when thinking about people moving across international borders, resettling, and acquiring certain rights, such as access to free public education, work rights, welfare provisions, residency rights, voting rights, and political protection and representation under the law. Mainstream migration and citizenship perspectives assume place-bound membership as primary, and movement between social points as secondary. Under this logic, people who move—and attempt to move—across international borders are perceived by nation-states and associated guardians of mobility as "secondary or derivative figures with respect to place-bound social membership" (Nail 2015, p. 3). When research uncritically accepts the premise that human mobility is an activity that is naturally subject to the authority of nation-states and their associated border and

citizenship regimes, the findings produced naturalize the nation-state as being the legitimate agent of inclusion and exclusion, while leaving unexamined the power dynamics involved in producing these highly uneven and often contested relationships.

When conceptualizing human mobility through a traditional migration and citizenship lens, static place (the nation-state) and membership (citizenship) are theorized first, followed by the 'migrant', who is conceptualized as lacking both stasis and membership. Under this logic, border crossing mobile people are perceived to be questionable, or risky, at best and threatening, or potential terrorists, at worst. People with 'illegal migrant' status are imagined as lacking not only status and membership, but also lacking the 'legitimate' right to be physically present in a nation-state. As De Genova (2007) rightly observes, "migrant 'illegality,' however, like citizenship itself, is a juridical status. It signifies a social relation to the state; as such migrant 'illegality' entails the production of a pre-eminently *political* identity" (p. 425).

Leaning upon the nation-state construct, people are sorted and classified into various migrant and citizenship statuses, including 'skilled migrant,' 'temporary migrant,' 'permanent resident,' 'refugee,' 'asylum seeker,' 'citizen,' 'non-citizen,' and so on. As these categories are applied to people, they become constructed into various imagined groups, whose statuses are often codified into citizenship and migration legislation. The result is to naturalize legal statuses onto the bodies of people by referring to them as 'citizens,' 'non-citizens,' and so on whose corresponding rights and responsibilities to the nation-state are based upon their citizenship and migration status. In other words, the nation-state is taken as the obvious guardian of who shall (and who shall not) traverse its borders and access various social, economic, and human rights within its borders and citizenship and migrant status are naturalized as the obvious conditions upon which certain people have rights (and certain others do not). Through this widespread nation-state ideology, which is largely perceived as non-ideological, a "ubiquitous division [is] enacted between the more or less 'rightful' members (citizens) and the relatively rightless non-members (aliens)" (De Genova and Peutz 2010, p. 7), or 'non-citizens.' Across many nation-states, immigration authorities have devised more complex bureaucratic and administrative instruments to determine who has the 'legitimate' right to enter a nation-state's territory and what criteria 'non-citizens' must satisfy to access migration and citizenship pathways that allow them to continue to reside, work, and become eligible for welfare and voting rights.

Citizenship is itself an "idea of inclusion [that] relentlessly produces exclusion" (Isin 2005, p. 381). Most migration-oriented research "is still characterized by an underlying assumption that everyone should somehow 'have' citizenship in order to be 'social' and thus receive social/state security' (Raithelhuber et al. 2018, p. 7). Migration scholars, including Raithelhuber et al. (2018) and Bauder (2013), have suggested that even the more critical contributions to migration, citizenship, and border studies are often ideologically contained within a perspective that continues to embrace citizenship. Critical citizenship scholarship often explores how "understandings of citizenship are forged and struggles for full citizenship are waged" (Lister 2007) by excluded groups in various contexts. These studies often attempt to extend the category of citizenship, such as through notions of "inclusive citizenship" (Lister 2007; Kebeer 2005). Certainly, this literature makes valuable contributions by casting a more critical eye on dominant understandings of citizenship and providing tools for marginalized groups to access greater social justice within the confines of the current hegemonic system of nation statehood, citizenship, and border regimes. However, these studies often continue to remain institutionally captured within these dominant paradigms and unwittingly continue to leave the hegemony of international system of nation-states uninterrogated.

The value of being categorized as 'a citizen' rather than 'a non-citizen' is that "the exercise of virtually all rights depends on territorial presence within the state, and only citizens have an unqualified right to enter and remain on state territory" (Macklin and Bauböck 2015, p. 2). At best, 'non-citizens' are treated as 'eternal guests' (Kanstroom 2007) of the nation-state, whose "residence is contingent on a certain standard of behavior and/or adherence to immigration laws" (Anderson et al. 2011, p. 549).

In recent years some nation-states have, or are considering, legislating considerable changes to their immigration and citizenship laws, policies, and practices. Increasingly, these changes to immigration, citizenship, and border protection are portrayed as being 'hard-line' or tough on migrants. As nation-states increasingly adopt hard-line approaches towards those perceived as outsiders, immigration and citizenship policies and practices are reengineered ways that increase the potential for people to become 'non-citizens' and be subjected to deportation and detention powers. To illustrate this point, in 2015 Australia passed legislation that made it possible for nearly half of all Australians to lose their citizenship if they act "inconsistently with their allegiance to Australia" (Australian Citizenship Amendment Act 2015). This broadly-worded legislation has the potential to affect more than 10 million people in the country who have dual citizenship or are 'foreign nationals,' i.e., people with citizenship status who were born overseas.

## 6. Pressing for Sub-Citizen Solidarity

This paper has explored the experiences of children held in immigration detention in Australia and the United States. However, it should be reiterated that sub-citizenship does not only apply to those in immigration detention. As all people live within the international nation-state system and market fundamentalism continues to expand and intensify, it follows that processes of sub-citizenship affect 'citizens' and 'non-citizens' around the world, but in highly uneven ways. Since the neoliberal globalization paradigm began in the 1980s, nation-states regularly reengineer citizenship, immigration, and border regimes in ways that align with capital accumulation. The result is that 'citizens,' and especially 'non-citizens,' tend to experience these restructures through increased levels of precarity and subordination with respect to nearly all aspects of life, including safety, education, employment, political representation, welfare entitlements, residency, and personal liberty.

As discussed above, neoliberal statecraft works by harnessing the state to reengineer the pathways leading toward citizenship as well as the conditions and tenor of citizenship status to further the goals of market expansion and capital accumulation. On the ground this means all people, regardless of their citizenship status, are increasingly positioned to 'feel the pinch' of market citizenship. Neoliberal statecraft works against the mobility, security, and human rights of people as it reengineers citizenship and migration labels and rules to serve the interest of capital accumulation and market expansion. With respect to people attempting to move away from conditions of human and economic insecurity, statecrafting immigration and border regimes involves intentionally slowing down and disconnecting the movement of people with 'illegal migrant' status. As nation-states harness people into the privatized immigration detention industry capital accumulates, markets expand and violence is directed at children and adults construed as 'illegal.'

Going forward, I suggest that a citizenship/sub-citizen lens could further the goals of migration and mobilities researchers who are interested to investigate how it has come to be that certain groups of people experience various kinds of (im)mobilities and differential access to employment, education, safety, and human rights. More generally, I suggest that the notion of sub-citizenship can be used to bring forth and explicate the connections between neoliberal governance and the political crafting of various kinds of precarity and social inequalities, such as Sparke has shown by demonstrating the link between austerity and people's experiences of ill-health.

Processes of sub-citizenship are primed to expand and intensify with neoliberal reengineering of all aspects of life. As sub-citizenship expands, more people will experience heighted precarity and some will experience extreme forms of dehumanization characterized by various kinds of human rights violations. In many Western states, we can clearly see the intensification of sub-citizenship happening as immigration pathways are increasingly 'skills based' and commodified; citizenship pathways are becoming longer and more contingent on certain codes of behavior; and border regimes are becoming more militarized, privatized, profitable, expansive, and abusive to people deemed to be deportable, detainable, and, increasingly, torturable by the nation-state. These developments should

not be considered as isolated events. Rather, they are structurally linked to the internal neoliberal globalization and the mantra of indefinitely expanding markets.

If there is any hope of slowing down or reversing processes of sub-citizenship, I think it must lie in the recognition that, under the neoliberal globalization paradigm, the state will continue to redraw the tenor of citizenship in ways that align with markets but often work against human mobility and human rights. If this trend continues unabated, sub-citizenship is likely to intensify, expand, and produce greater violence and precarity for 'non-citizens' and for those with more secure 'citizen' status. This article has drawn upon contemporary high-profile cases of children in immigration detention to illustrate what sub-citizenship can look like for those positioned on lower rungs of the sub-citizen hierarchy. Future research is needed to explore people's diverse and localized experiences with processes of sub-citizenship. This scholarship will build greater understandings of what sub-citizenship looks and feels like for diverse groups of people with different citizenship and migrant statuses in various historical and temporal locations. There is much work for future research to explore and map out multiscalar processes of sub-citizenship that cut across and impact upon people through a range of intersections including citizenship and migrant status, socio-economic status, race, religion, gender, sexuality, age, ability, and other markers of difference.

Developing a greater understanding of what sub-citizenship is and how it shapes the everyday worlds of people around the world has the potential to raise radical new forms of solidarity. The concept of sub-citizenship can act as a powerful aid for reimagining entirely new possibilities for human mobility and access to social, economic, juridical, and political rights regardless of status. As we gain more knowledge about how people's lives are actually affected through processes of sub-citizenship, we can build recognition of how sub-citizenship operates and forge a kind of sub-citizen solidarity between 'citizens' and 'non-citizens' alike, regardless of their legal status. Perhaps with increased recognition and knowledge of how sub-citizenship functions to consolidate power and capital for the few at the cost of the many, the world's sub-citizens may begin to collectively call for genuine and universal access to mobility rights, an end to all forms of arbitrary detainment, and ultimately demand 'status for all.'

**Funding:** This research was funded by Charles Darwin University's International Postgraduate Research Scholarship, 2013–2017.

**Acknowledgments:** I conceived of the idea of processes of sub-citizenship during my PhD, which was completed in 2018. I wish to thank my supervisors, Brian Devlin, Georgie Nutton and Nici Barnes, for their support and the many discussions that helped to deepen and refine my thinking on this concept. I also wish to thank Ernesto Castañeda and three anonymous reviewers for their valuable critiques and suggestions to this paper.

**Conflicts of Interest:** The author declares no conflict of interest.

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
