# Peer review of "Processes of Sub-Citizenship: Neoliberal Statecrafting ‘Citizens,’ ‘Non-Citizens,’ and Detainable ‘Others’"

_socsci, doi:10.3390/socsci9010005_

Round 1
Reviewer 1 Report
This paper is highly relevant to the mission of the Social Sciences. It addresses an important topic that is directly related to international migration (migration of children), citizenship, human rights.
It is a very valuable discussion of an important topic.
I suggest to:
describe much more about the methods explore the literature related to human rights (part 2 and 3 of the paper) and migration of kids (part 4).Author Response
Please see the attachment.

Reviewer 2 Report
This paper analyses processes of securitisation and criminalization of migrants around the world, and seeks to identify discursive and legal mechanisms through which state violence is legitimised. It uses the concept of ‘sub-citizenship’ to explain how exclusion and violence is enabled through the dehumanization of citizens and non-citizens alike. It also argues that migration control mechanisms work hand in hand with nationalistic and neoliberal projects to exclude unwanted outsiders. It seeks to illustrate this process by considering recent policies and practices of detention of children in the USA and Australia.
While this paper provides a good analysis of the existing literature on the subject, it has a number of weaknesses. First, it is not clear what the contributions of the paper are as the author does not clearly explain what the concept of sub-citizenship adds to understanding contemporary processes of exclusion. While sub-citizenship addresses both citizens and non-citizens, the paper is clearly is about non-citizens so one wonders how does the concept connects to the remit of the paper? It is also surprising that the author does not mention race and racism in the analysis. Further, the author uses highly loaded concepts, such as ‘neoliberalism’, without explaining in greater detail how do they connect to the issue of migration. As such the analysis pulls on different directions without a proper wrap up.
Second, the theoretical and conceptual parts at the beginning and end of it are loosely connected to the ‘case studies’ on children detention. Further, it is not clearly explained why these cases are chosen and what their relevance is to the topic of the paper. Quotes and drawings are also improperly referenced and attributed, and I am slightly concerned about potential copyright issues arising from the material used from secondary sources.
Overall, I found this paper extremely long, unarticulated and methodologically problematic. Much of the analysis is a review of other sources which leaves the reader asking what the novelty and contributions to the field are.
Reviewer 3 Report
This article is an innovative approach to how we conceptualise citizenship. It is an exciting piece of work which offers important new theoretical tools for understanding the ordering of populations along contemporary, neoliberal lines. It argues that those who are most vulnerable to the loss of full citizenship, or blocked from attaining it at all, are those posed as risky or unproductive populations or on the margins/outside the nation. The author reveals the processes and practices which normalise the nation state as a common sense category while obscuring the violence of the hierarchies of personhood (citizenship) which it produces. The discussion on deportation powers is especially relevant and timely to highlight, given its increasing use by Australia to banish/exile those posed as risky, threatening or needy individuals.
The connection to inequality that is discussed in the paper is something which Emma Larking analyses in relation to human rights in her book Refugees and the Myth of Human Rights where she points to the limits of equality for refugees. This could be considered in later work, should the author wish to do so.
Other work which might be worth considering are: Bauman's work Wasted Lives and Anne McNevin's article on the paradox of neoliberalism (2007) Australian Journal of Political Science; Marrus The Unwanted on the history of human mobility as a key text in this area and Katja Franko on the globalisation of the border, especially in her work 'The Ordered and the Bordered Society' (2017).
A few key recommendations:
1. The author notes that this does not mean that 'illegal' migrants can even achieve sub citizenship, but that it is instead the potential categorisation which might lead to less secure citizenship. This is an important distinction which bears some further reflection and emphasis.
2. The discussion of the treatment of children in immigration detention in Australia and the US would be stronger if the author was more explicit about how such violence is possible. That is, that the violence this reveals is made possible by categories of the human by the nation state that renders this suffering merely as the collateral damage of border protection. Some analysis of this will also strengthen the link between this section and the remainder of the paper.
3. The author does not need to apologise for not addressing all categories of migration and can state this with greater confidence that the choice is to focus on specific categories because of what they reveal about the experiences of those at the lower levels of the hierarchy.
4. The figure denoting the hierarchy of citizenship is interesting but does not fully account for the experience of those who have citizenship but nonetheless are subject to processes of exclusion and marginalisation on account of, for example, class, racialisation, gender and sexuality etc. Accordingly it fails to adequately denote the actuality of limited citizenship for some.
I noted some typos which I am sure the proof editing will pick up, for example the spelling of hierarchical on page 2, and the statement 'we are all sub citizens on our way to becoming sub citizens'. p. 24 and wondered if this is an error.
